# Importance of Self-Efficacy in Eating Behavior and Physical Activity Change of Overweight and Non-Overweight Adolescent Girls Participating in Healthy Me: A Lifestyle Intervention with Mobile Technology

**DOI:** 10.3390/nu12072128

**Published:** 2020-07-17

**Authors:** Anna Dzielska, Joanna Mazur, Hanna Nałęcz, Anna Oblacińska, Anna Fijałkowska

**Affiliations:** 1Department of Child and Adolescent Health, Institute of Mother and Child, 17a Kasprzaka St., 01-211 Warsaw, Poland; j.mazur@cm.uz.zgora.pl (J.M.); hanna.nalecz@imid.med.pl (H.N.); anna.oblacinska@imid.med.pl (A.O.); 2Department of Humanization in Medicine and Sexology, University of Zielona Gora, Collegium Medicum, Energetykow St. 2, 65-729 Zielona Gora, Poland; 3Department of Cardiology, Institute of Mother and Child, 17a Kasprzaka St., 01-211 Warsaw, Poland; anna.fijalkowska@imid.med.pl

**Keywords:** healthy lifestyle intervention, school-based intervention, eating behavior, MVPA, overweight and obesity, self-efficacy, adolescent girls

## Abstract

Very little is known about how multicomponent interventions directed to entire populations work in selected groups of adolescents. The aim was to evaluate the effectiveness of the Healthy Me one-year program on changes in healthy eating and physical activity among overweight and non-overweight female students. Randomization involved the allocation of full, partial or null intervention. The randomized field trial was implemented in 48 secondary schools (clusters) all over Poland among 1198 15-year-old girls. In this study, a sample of N = 1111 girls who participated in each evaluation study was analyzed. Using multimedia technologies, efforts were made to improve health behaviors and increase self-efficacy. The main outcome was a health behavior index (HBI), built on the basis of six nutritional indicators and one related to physical activity. HBI was analyzed before and immediately after intervention and at three months’ follow-up, and the HBI change was modeled. Statistical analysis included nonparametric tests and generalized linear models with two-way interactions. Comparing the first and third surveys, in the overweight girls, the HBI index improved by 0.348 (SD = 3.17), while in the non-overweight girls it had worsened. After adjusting for other factors, a significant interaction between body weight status and level of self-efficacy as predictors of HBI changes was confirmed. The program turned out to be more beneficial for overweight girls.

## 1. Introduction

According to the World Health Organization (WHO), adolescence starts in the second decade of life [1]. This period requires special attention because of its specific health and developmental needs and rights [2]. During adolescence, the transition period from childhood to adulthood, health behaviors are shaped and consolidated. Therefore, a healthy lifestyle is crucial for adolescents’ proper growth and development. Moreover, targeting adolescents with health behavior-shaping intervention activities affects the burden of disease in adulthood, providing better health through the ripple effect [3,4].

Nearly 40 years of the cross-sectional Health Behavior in School-aged Children (HBSC) study has consistently identified burning problems and the most vulnerable groups of adolescents in the European region of WHO and Canada [5]. The comparison of health behaviors of adolescents of both sexes indicated a co-occurrence of positive trends in boys and negative in girls. That resulted in an elimination of gender-related differences in the frequency of many negative behaviors [6] and exposed the population of 15-year-old girls—the future mothers of the next generations—as extremely vulnerable, especially in the context of persistent disadvantages in girls’ self-rated health, observed in many countries.

According to the international report from the HBSC study [5], obesity or overweight was found in 14% of 15-year-old girls and 36% consider themselves too fat. Moreover, girls aged 15 do not regularly eat breakfast on school days (52%) and do not eat fruit (62%) and vegetables (61%) every day, but every day they eat sweets (28%) and drink sweet carbonated drinks (15%). In addition, only 11% of them meet the recommendations for appropriate levels of moderate-to-vigorous physical activity.

Both systematic reviews of intervention programs [7] and guides for the prevention of obesity in children and adolescents [8] indicate the limited effectiveness of obesity prevention programs. Low effectiveness of these programs was found in children under 12 years of age and the introduction of interventions in young people aged 13–18 did not contribute to reducing BMI. Unfortunately, there is little research in this age group. Hence, it is difficult to give a reliable assessment of the effectiveness of the intervention [9].

Health-related behaviors are correlated, and many different consolidated patterns of behaviors can be observed in different environments [10]. Systematic reviews confirm that interventions aimed at improvement in moderate-to-vigorous physical activity have a simultaneous effect on empowering other health-related behaviors such as healthy eating or weight management [11]. Results of meta-analyses show that school-based interventions including a combination of healthy eating and physical activity may prevent overweight in the longer term [12] and also indicate moderate effectiveness of educational interventions in improving eating behaviors and ambiguous results concerning anthropometric changes [13].

Likewise, better intervention outcomes are associated with long-term interventions [14], as well as with the inclusion of a higher number of applied behavioral change techniques [15,16,17]. Incorporating behavioral change techniques focused on self-regulation into the intervention was found effective in changing physical activity and eating behaviors. Avery et al. (2012) confirmed this relationship in adult studies [18] and Martyn-Nemeth et al. (2009) in adolescents [19]. Furthermore, some studies demonstrate the effectiveness of interventions using interactive modern media to improve diet and physical activity of adolescents, although only a few indicate maintenance of the effect in the long term [20].

Effective behavior change requires the acquisition of appropriate skills that will allow activities to be initiated consistent with acquired knowledge. Moreover, it is extremely sensitive to environmental context [21]. One of the personal competences necessary to successfully implement changes in health behavior is self-efficacy, which has a proven link to motivation, behavior control and goal achievement [22]. By being convinced of one’s own effectiveness, a person gains the ability to initiate and continue changes even when faced with emerging challenges [23].

To date, the assessment of the effectiveness of the Healthy Me program has been carried out in the whole study group, without distinguishing between girls with and without excess body weight [24]. The implemented program was a universal prevention aimed at the whole population of 15-year-old girls. In the reviews of systematic community obesity prevention programs, reducing the prevalence of obesity is often assumed to be the main outcome [25]. Less attention is paid to assessing the changes in health behavior of students with and without excess body weight. However, the question arises—to what extent do overweight teenagers use universal programs? Is it a group representing less advantageous health behaviors, and do any beneficial effects of the program remain in this group after its completion? The presented paper fills this knowledge gap, while at the same time providing a picture of the effectiveness of this innovative program, which tried to reach its addressees with the use of modern multimedia technologies.

The aim of the study was to evaluate the effectiveness of the Healthy Me intervention program on changes in the prevalence of healthy eating behaviors and the level of physical activity among 15-year-old girls in Poland. It has been hypothesized that the effectiveness of an intervention may differ in overweight and non-overweight girls, and the improvement of personal competence may be a factor strengthening the effectiveness of the intervention [26]. Therefore, the main issue was to determine in which groups of girls the Health Behavior Index (HBI), consisted of seven indicators of eating behaviors and physical activity, improved taking into consideration their body weight status, change in self-efficacy, the type of intervention provided and possible effect of school environment.

## 2. Materials and Methods

### 2.1. Study and Intervention Design

The data were obtained from the randomized field trial with cluster randomization by school and repeated measures. In total, 1198 15-year-old girls, from 48 randomly selected secondary schools all over Poland, participated in the one-year Healthy Me program in 2017–2018. Schools were randomly assigned to the subsequent groups: full intervention group (24 schools, 636 girls), partial intervention group (12 schools, 277 girls) and null intervention group (12 schools, 285 girls) (Figure 1).

The main area of interest was the improvement in physical activity, although the intervention activities were conducted in four thematic phases: physical activity, eating behavior, risk behavior and personal and social competencies. The multicomponent intervention used mobile technology (a dedicated mobile application and a fitness band) and involved a combination of techniques. The Healthy Me program used Social Cognitive Theory [27] as its theoretical foundation and was based on an interactive technology approach [26]. The intervention included behavioral and environmental components. Self-efficacy was shaped by setting goals, observing others and receiving feedback from the technologies (fitness band, app) that supported self-monitoring. However, the type of intervention depended on the type of intervention group, which made it possible to assess the effectiveness of particular sets of intervention methods and techniques (Table 1).

The study and the intervention procedure were accepted by the Bioethics Committee of the Mother and Child Institute in Poland (number: 32/2017 from 22 June 2017) and the funding body (Ministry of Health in Poland, Grant no. 6/7/K/6/NPZ/2017/106/622).

### 2.2. Evaluation Surveys

The project has been fully evaluated, and as part of the evaluation of the intervention results, questionnaire surveys were conducted three times during the project implementation:Study 1—at the beginning of the program implementation (November 2017).Study 2—after the intervention completion (June 2018).Study 3—three months after the intervention completion (September 2018).

Each questionnaire contained a similar set of questions to allow comparisons to be made about changes in subjective health, different health-related behaviors and related factors.

Anthropometric measurements (e.g., weight, height) were conducted three times by school nurses, once in each survey round.

### 2.3. Sample Characteristics

The present analyses cover girls (N = 1111) who have completed three rounds of the survey (Table 2). About half of the girls participated in the full intervention group, and half belonged to the partial and null intervention groups. Based on the WHO standards [28], almost a quarter of participants were assessed as overweight or obese (23.5%), and the frequency was higher than in the groups of similar age form cross-sectional HBSC, probably due to the different cut-off point used for the estimation of body weight status [6]. The percentage of BMI missing data in the studied sample was very low (0.8%). A similar percentage of girls with excess body weight occurred in each type of intervention group. At the baseline, the overweight and non-overweight groups did not differ in terms of the scores of the HBI or the general index of self-efficacy (GSE), both described below.

### 2.4. Measures

#### 2.4.1. Health Behavior Indicators

Six indicators related to eating behaviors and one measure of physical activity were tested in these analyses.

1.Eating behaviorsFrequency of eating fruits, vegetables, sweets, drinking soft drinks with added sugar. Girls answered how often they eat or drink the products by choosing one answer from seven categories, from “never” to “daily, more than once”.Breakfast consumption. Girls were asked to answer the questions on the frequency of eating breakfast on schooldays, choosing from six answer categories, from “never” to “five days a week”, and during the weekends, choosing from three options, from “never” to “both days”.2.Physical activityModerate-to-vigorous physical activity. Girls answered the question: “Over the past seven days, on how many days were you physically active for a total of at least 60 min per day? Please add up all the time you spent in physical activity each day”. The questions had eight response categories: from “zero days” to “seven days”.

The frequency distribution of girls undertaking the above-mentioned eating behaviors in subsequent study periods, by type of intervention and body weight status, is presented in the Appendix A. The above questions come from the HBSC study protocols and have been tested repeatedly in Poland in a population similar in age [6].

#### 2.4.2. HBI

The summary HBI was estimated for all three study periods. The index consists of seven indicators of eating and physical activity behaviors mentioned above. The response categories in each behavior were recoded and scored from 0 to 3 points, as follows, with a higher value indicating a more favorable result:Fruit and vegetables: 0 points—never or less than once a week; 1 point—“once a week”; 2 points—“two to six days a week”; 3 points—“daily, at least once or daily more than once”.Sweets and soft drinks consumption: 0 points—“daily, at least once or daily more than once”; 1 point—“two to six days a week”; 2 points—“once a week”; 3 points—“never or less than once a week”.Breakfast consumption on schooldays: 0 points—“never”; 1 point—“one to two days”; 2 points—“three to four days”; 3 points—“daily”.Breakfast consumption on weekends: 0 points—“never”; 1 point—“one day”; 3 points—“both days”.Moderate-to-vigorous physical activity: 0 points—“zero days”; 1 point—“one to three days”; 2 points—“four to six days”; 3 points—“seven days”.

The highest value (3 points) attributed to the recoded answers to the above questions was consistent with the national recommendations on the frequency of eating different groups of products and meals [29], as well as the global moderate-to-vigorous physical activity guidelines for children and adolescents [30].

The summary score of the HBI was from 0 to 21 points. HBI scores in each of the three evaluation surveys are presented in Table 3 and Table 4.

In building the HBI, its six different variants were considered. Some factors were excluded, and attempts were made to additionally include intense physical activity and meals eaten together with parents. The psychometric properties of individual indices in three study periods and the significance of the level of their changes were evaluated. None of the analysed indices had a single factor structure, and the internal consistency was slightly below the recommended level of 0.70 which is accepted for larger sample analyses [31]. The advantage of the chosen index is the fact that it takes into account the level of physical activity, which was a key element of the intervention. Eating healthy food most strongly affects the variability of the selected index. Eating sweets appeared to be the weakest component. However, this element was not abandoned, due to a considerable decrease in the frequency of eating sweets during the project implementation period (Appendix A). There was only one case of missing data in the HBI (n = 1).

#### 2.4.3. Self-Efficacy—Personal Competence Scale

To measure the change in self-efficacy the KompOs scale was used. This is a two-dimension, 12-item, standardized questionnaire by Z. Juczynski, applied for younger and older adolescents to assess their self-efficacy [32]. In older adolescents (15–17 years) this tool has a two-dimensional structure and measures strength to initiate behavior and perseverance to sustain it. Psychometric analysis performed on our sample at the baseline revealed good reliability of the full scale, with Cronbach’s *α* = 0.757, as well as the component scales: for strength Cronbach’s *α* = 0.736 and for perseverance Cronbach’s *α* = 0.677. In other studies, test–retest reliability of the scale, applied in older adolescents, was 0.51. The theoretical validity of the scale was tested and showed a positive correlation with General Self Efficacy [33] *r* = 0.43 and Coopersmith Self-Esteem Inventory (CSEI) [34,35] *r* = 0.30.

In the following description, instead of the national scale abbreviation (KompOs), the term self-efficacy is used. The general self-efficacy score (GSE), as well as two partial scores of strength and perseverance, were analyzed. The percentage of missing data in GSE was 6.6% and 6.2% in the first and third study, respectively.

#### 2.4.4. Body Weight Status

Results from the anthropometric measurements (body weight, height) conducted by school nurses before the intervention (November 2017) were used. BMI classification was made using WHO standards [28]. For the analysis, the BMI variable was recoded into two categories of body weight status: (1) overweight (overweight and obese categories) and (2) non-overweight (other categories).

### 2.5. Statistical Analysis

A combined analysis of independent and dependent observations resulting from repeated measurements, which is an approach commonly used in the case of mixed data, was applied.

The HBI changes constituted the main outcome variable. They were analyzed by comparing successive measurements and examining the determinants of the changes themselves, which only required the technique of comparing independent samples. The most important variable was the HBI change between the first study and follow-up three months after intervention, because of simultaneous measurements of competence at these time points.

Due to the non-normal distribution of the HBI values and the HBI changes, non-parametric methods were used for two (BMI groups) and three (types of interventions) adolescent girls’ groups, respectively. These were Wilcoxon and Kendall tests for dependent data and Mann–Whitney and Kruskal–Wallis tests for independent data.

The school effect was also examined by estimating the ICC (intraclass correlation coefficient). A mixed linear model with school as a random effect was used for this purpose. The ICC values for different types of interventions were compared separately for the absolute value of the HBI and the changes in this index.

In a multifactor analysis, a generalized linear model was estimated (GENLIN procedure in IBM SPSS software, v.23). It is a method that does not impose strict conditions as to the distribution of the analyzed variables, allowing various types of variables to be included as predictors (binary, categorical, continuous) and enabling a transparent analysis of the interaction effect.

Three GENLIN models were estimated, describing the determinants of the HBI change on the basis of the results of the Study 1 and Study 2, Study 2 and Study 3, and Study 1 and Study 3 evaluation surveys. After checking variants of the models, it was decided to include in the group of predictors the following: body weight status, the type of intervention and the interaction between the body weight status category and the change in self-efficacy. The analyses of the HBI change were also corrected with respect to the initial HBI level and the self-efficacy score. The overall quality of the models was measured by the omnibus test. It gives the answer to the question whether the explained variance in a set of data is significantly greater than the unexplained variance.

## 3. Results

### 3.1. HBI

The mean scores of the HBI in all three study periods in the overall sample, by body weight status are presented in Table 3, and by the type of intervention group in Appendix A.

The HBI in Study 1 did not differ by body weight status. In Study 2, it was slightly higher in overweight than non-overweight girls, but the results were at the tendency to significance level (*p* = 0.052). Three months after the intervention (Study 3), the overweight girls presented significantly higher scores of HBI than non-overweight girls (*p* < 0.01). The highest HBI scores were indicated in the full and null intervention groups in all three study rounds, while the lowest occurred in the partial intervention group.

In the total sample, as well as in both groups distinguished by body weight status, significant differences were found in the HBI scores between the three rounds of the study. Comparing the initial level and results three months after the Healthy Me program completion, the crude level of change in HBI was equal to 0.026 (SD = 2.89). In the group of girls with overweight or obesity, an improvement was observed (0.348 ± 3.17), while in girls without excessive body mass health behaviors worsened.

### 3.2. Self-Efficacy—Personal Competence

Table 4 compares the distributions of self-efficacy indices, taking into account two available measurements, at the beginning of the Healthy Me program implementation (Study 1) and at follow-up after three months (Study 3). A decreasing trend in GSE was observed, which was caused by a considerably deteriorating assessment of the strength dimension, with slight changes in the level of perseverance. Unfavorable changes were observed only in non-overweight girls. In the overweight or obese group, changes in the general index and sub-indices were not statistically significant. These two groups of girls distinguished by body weight status did not differ considerably with regards to the general score, as well as regardless two dimensions of self-efficacy scale, both at the beginning of the program and three months after its completion.

Appendix A compares the results of non-parametric tests of the distribution of GSE, as well as the domains’ scores in the three intervention groups. At the onset, the girls from the schools covered by full intervention achieved the best results, while in the control group (null intervention) those indices were the lowest. Observable differences concerned only Study 1, GSE and the dimension of strength. The third measurement point (three months’ follow-up) did not reveal any significant differences between the intervention groups. Comparing the level of self-efficacy change in conjunction with the paired data test, a significant deterioration was shown in the full intervention group, which also concerned the overall score and the dimension of strength. A clear trend of a deterioration in competence level was also found with respect to the dimension of perseverance in the partial intervention group.

In reference to the initial hypothesis, HBI changes were checked depending on the level of GSE changes. It was contractually assumed that the deterioration and improvement would occur in case of a change by more than two points. In the three groups representing worsening, lack of change and improvement in GSE, there were 32.5%, 39.9% and 27.6% of girls, respectively. The percentage of girls with improved GSE was 28.4% in the overweight group and 27.0% in the non-overweight group (*p* = 0.792). According to the data presented in Figure 2, a significant change in GSE is associated with an improvement in dietary behavior and physical activity, measured by change in HBI. The impact of improved self-efficacy is more evident in overweight girls. In this group, even with a GSE change around zero, a slight improvement in HBI values has already been noted.

### 3.3. School Effect

Taking the HBI change between the results of the first and the third evaluation study as the most important outcome, it was examined to what extent this change depends on local school conditions. The ICC index was calculated. In the whole sample of 48 schools, it equaled 0.012. For particular types of intervention, it was estimated at the level 0.006 (full), 0.020 (partial) and 0.015 (null). This means that the proportion of variance in the HBI change that lies between schools is very small and slightly varies depending on the type of intervention. At the same time, the low ICC value allows for the abandonment of multilevel analyses taking into account the hierarchical data structure.

For comparison, the school effect in the whole study group and in relation to the absolute HBI value at the onset (Study 1) equaled 0.031 and 0.044 in Study 1, and 0.039 in Study 3. An increase in the ICC may be a signal that schools were not implementing the intervention program to an equal extent over the entire duration of the program.

### 3.4. Independent Predictors of the Change in HBI

Table 5 shows the results of the estimation of generalized linear models in which the dependent variable was the HBI change, calculated on the basis of the results of different surveys (1 and 2; 2 and 3; 1 and 3).

The models were adjusted to the initial levels of the HBI and the self-efficacy score. The selected predictors accurately described the fluctuations of the HBI changes between first and second measurement points and between first and third points (deferred effect). The middle model (2 and 3) described to a small extent the determinants of the HBI changes immediately after the end of the program. The overweight girls achieved significantly higher HBI gains compared to peers without excess body weight in both extreme models (Table 5). The intervention effect was best demonstrated in the last model, describing the change between the first and third surveys. In the case of partial intervention, the changes were less beneficial. A significant interaction between the changes in self-efficacy and body weight was also shown. In the first and third models, among girls with excess body weight, the improvement in personal competence contributes more to the increase in the HBI value. For example, when comparing the first and third measurement points, an increase in the self-efficacy by one unit results in an increase in the HBI by 0.132 (*p* = 0.006) in the overweight and obese group. In the group of non-overweight girls, the HBI increase was only 0.037, and this parameter of regression function does not differ significantly from zero (*p* = 0.198).

On the basis of the above three models of the HBI change determinants, it is possible to estimate the theoretical values at the second and third measurement points in two groups of girls with different body weight statuses, starting from the actual initial value (Figure 3).

In both groups, an increase in the HBI was observed between the beginning and the end of the Healthy Me program, followed by a decrease, according to the measurement three months after the end of the program (Study 3). This initial improvement in health behaviors was clearly greater in the overweight group. Comparing the first and third measurement points, it is possible to draw a conclusion regarding the effectiveness of the program as a tool for improving health behaviors. In the group of girls without overweight or obesity, the effectiveness of the program is lower, and extreme measurements indicate a return to the baseline and even a slight deterioration in the HBI. Attempts to devise alternative models have not led to better results. Among other things, the independent influence of partial indices of self-efficacy (strength and perseverance) was studied, and attempts were made to include the main effect of self-efficacy in the model. The model that takes into account the interaction of body weight status with self-efficacy was considered optimal.

## 4. Discussion

In our research, we assumed that a change in the level of HBI may be related to a change in the level of personal competence among girls participating in the Healthy Me intervention program, which used mobile technologies. Our analyses confirmed this assumption. The change in the HBI was explained by the interaction of the self-efficacy level with body weight status. Although there was no positive change in health behavior among girls without excess weight, girls with excess body weight (overweight or obesity) achieved a better score in the health behavior index in the follow-up study after three months of intervention.

Prior to the intervention, there were no observable differences in the values of health behavior indices between overweight and non-overweight adolescent girls. Taking into account the type of intervention, slightly lower values at the starting point occurred in the partial group than in the other two intervention groups. The other studies conducted among Polish schoolchildren also support our conclusion that maintaining a diet rich in beneficial products is not the domain of adolescents without overweight or obesity and even more often occurs in overweight or obese adolescents [36]. Conversely, some problems are more frequently observed in overweight teenagers compared to their peers without excess body weight, such as skipping meals [37], having fewer meals during the day [38] and lower physical activity [39,40].

The level of the HBI changed in the second evaluation study, but after three months from the end of the intervention, it returned to a level close to the initial level. It turns out that in girls without overweight or obesity, a slightly lower HBI score between extreme measurements was recorded, but in the group of girls with excess body weight, an improvement was observed. Moreover, the deferred effect revealed a significant difference in the average HBI indices in favor of overweight and obese teenagers. The result seems all the more interesting because our intervention was not aimed at changing behaviors of adolescents at risk—overweight, but at the general population of 15-year-old girls, which was selected because of the significant deterioration in health behavior for this age group. The aim of the program was to assess if the proposed intervention could help to slow down the unfavorable trend before they reached the age of 15. As effects were observed in the group of girls with excess body weight, it may be hypothesized that even if the study was not addressed to the adolescents from risk groups (selected prevention), the overweight participants may be more motivated to engage in prevention programs, which makes this group more vulnerable to benefits [41]. Based on our studies and thesis supported by other researchers, there is a need to cautiously draw conclusions about changes in health behavior induced by intervention, especially in the case of long-term programs carried out in the developmental period [42]. Among others, the negative changes resulting from developmental factors should be taken into account. Moreover, in a longer program, the rate of change could be altered by a number of interim measurements, promotion of the program before it starts, overlapping of other parallel trends or a negative effect of withdrawal from the program. Thus, the absence of a significant, positive change in the HBI in the general population of 15-year-old girls can be considered as a satisfactory result, taking into account developmental considerations, and a negative trend in health behaviors which increase with age, observed in other studies among girls [43].

The analysis of self-efficacy showed no differences among intervention types, nor the body weight status among the studied population at the starting point of the intervention. The main changes revealing the impact of the Healthy Me program concerned the general score and the self-efficacy dimension of strength (to initiate behavior) and showed the decrease in these scores among participants of the full intervention group. Within the partial intervention group, the dimension of perseverance also deteriorated, and this result supports the claim that the multicomponent, but moderate, intervention impacts have an exceptional effect on the participants’ conviction about the possibility of sustaining behaviors. This result might be caused by an ongoing dynamic process of verification of self-efficacy during the program.

Based on Social Cognitive Theory, the self-efficacy building strategy is one of the most effective tools in the health behavior change programs targeting diet and/or physical activity among children [44]. Jacobson and Mazurek Melnyk (2011) concluded after their pilot study with overweight and obese school-age children that healthy lifestyle interventions that include cognitive behavior skill building may be the key to strengthening the child’s healthy beliefs and facilitating healthy lifestyle choices and behaviors [45]. Morano et al. (2016) recommend that childhood obesity programs should target psychosocial correlates of physical activity [46], among which the crucial one, as Kołoło et al. (2010) indicate, is self-efficacy [47]. Higher self-efficacy is related to better decision-making and goal achievement [22]. Therefore, girls assessing their self-efficacy well are more likely to undertake (strength dimension of self-efficacy) and sustain (perseverance dimension) the health behaviors.

Our study shows significant interaction between self-efficacy and body weight. Among overweight girls, improvement in self-efficacy resulted in enhancement of health behaviors. The sense of interaction and the mediating role of self-efficacy and other social competences in the process of changing health behaviors is strongly established and widely proved in the literature. Especially regarding overweight and obesity, according to Goffman’s spoiled identity theory [48], and further randomized control studies of the stigma effect on health behaviors [49], children with low social competence are at higher risk for obesogenic behaviors. That interaction was also confirmed in a national sample of Americans where nine-year-old children with lower social competence were at higher risk of becoming overweight or obese by age 11 [50]. In low social competence groups, avoiding stress caused by complex psychosocial factors with negative feedback related to excess body weight may manifest in unhealthy behaviors such as solitary, sedentary or unhealthy eating. According to Melnyk et al. (2009), psychosocial factors may inhibit or cause barriers to healthy behaviors in adolescents [51]. On the other hand, Vila et al. (2004), using a Child Behavior Checklist, found that obese adolescents demonstrated significantly poorer social skills [52]. These studies show the nature of the reciprocal relationship among competences, body weight and health behaviors.

Additionally, the school effect was measured in the analysis. Based on results obtained, the effect of the school has proved to be small, indicating quite a consistent approach by schools towards the implementation of intervention activities. Interventions to improve health behavior are largely implemented in the school environment, and many of them have a positive impact on nutrition and physical activity [53]. Due to the availability of target groups as well as methods, resources and qualified staff, the school seems to be an ideal environment for health promotion and education [54]. However, a lot depends on the quality of the proposed interventions, the way they are implemented, consistency of the activities [55], proper preparation of the contractors, financial possibilities and the duration of the intervention [54,56]. In this regard, the small school effect obtained in our study may be the result of how schools were prepared to implement the intervention activities. Preparations included providing clear instructions, training of direct executors, application of unified educational methods and contents and a strictly determined sequence of undertaken activities. Moreover, the high competences of physical education teachers responsible for coordination and carrying out activities at schools may have had an impact on the uniform implementation of the program at the school-setting.

### Strengths and Limitations

We are aware of some limitations of our study, which were partly due to the schedule of the Healthy Me program, as well as the assumptions adopted in this article. First, the final deferred effect of the program should be assessed in the long term. Second, one of the most important variables, i.e., the level of self-efficacy, was not measured just after the end of the program but three months after the end of the intervention. Only a few factors were considered in the analyses, focusing on differences in the level of the HBI and its changes in the groups of overweight and non-overweight girls. The Healthy Me program was implemented in a variety of environments (48 schools all over Poland), over a long period of time (a year), covering a large group of girls (*n* = 1111). This environmental variability was undoubtedly an asset but also created additional limitations. In such a large and diverse group, it was difficult to control the involvement of individual schools in the implementation of the program, and the distinct differences are evidenced by the results of qualitative studies and different subjective evaluations of the program by its participants [24]. However, the low ICC values in this study may indicate quite consistent implementation of the intervention actions by the schools involved. It was not possible to analyze in detail the changes in the diet of the program participants. The main outcome variable, i.e., the HBI, contained a strong nutritional component but was corrected with respect to the level of physical activity. This version of HBI was chosen because improvement in physical activity was the main focus of the Healthy Me intervention.

Despite the above limitations, the analyses presented have a number of advantages and bring additional knowledge to research on the evaluation of multifaceted intervention programs. Attention was paid to the heterogeneity of the intervention group. Commonly, it is hypothesized that different intervention components will benefit equally different subgroups of participants in the hope of offering something for everyone. It has been proven that girls with excess body weight have benefited more from participation in the Healthy Me program, which is the main conclusion of these analyses. This was partly due to the change in their self-efficacy, which was at a relatively low level, but any improvement resulted in better health behaviors. Taking into account the aspect of personal competences is one of the strengths of this program. Self-efficacy was measured with a robust tool dedicated to the adolescents. Usually, the motivational and strengthening factors are only mentioned as a theoretical basis for intervention. In our program, this factor was one of the components to be evaluated. In addition, we have introduced an interaction effect into statistical analyses, which is now considered an important part of the search for an optimal intervention model [57].

## 5. Conclusions

In summary, our results demonstrate a significant effect of self-efficacy with the interaction of body weight status on improvement in eating behavior and physical activity among adolescent girls. The authors conclude that the positive impact of the intervention proved to be stronger for overweight girls. Girls with excess body weight, three months after intervention completion, presented a higher level of favorable health behaviors than girls without excess body weight. Further work is certainly required to disentangle these complexities in non-direct effects of interventions on health behavior change among adolescent girls. When analyzing the effects of such programs, it is necessary to take into account the multiplicity of interrelationships between different factors that may modify the effects obtained. Our paper opens new conceptual and practical fields in research on the evaluation of health interventions. Firstly, effective interventions targeting adolescent girls should include a strengthened element of developing personal competences, the growth of which appears to be most beneficial to girls at risk. Secondly, the level of change in personal competences should be monitored during the whole evaluation process. This seems to be far beyond including psychological factors as the only theoretical basis for the intervention.

## Figures and Tables

**Figure 1 nutrients-12-02128-f001:**
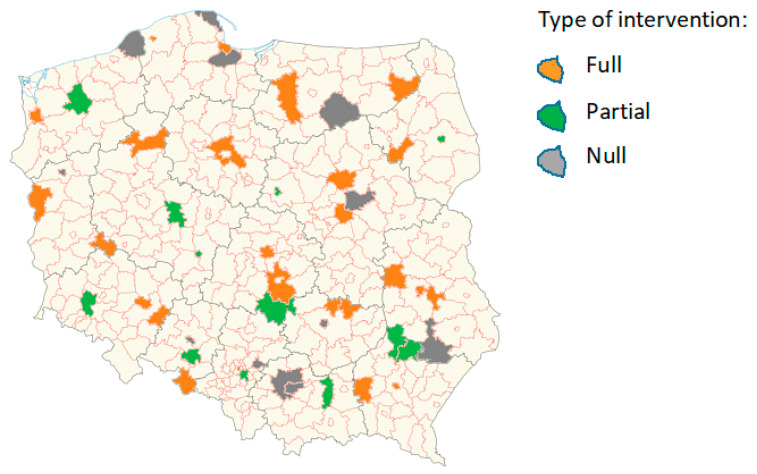
Location of schools participating in the Healthy Me program by the type of intervention.

**Figure 2 nutrients-12-02128-f002:**
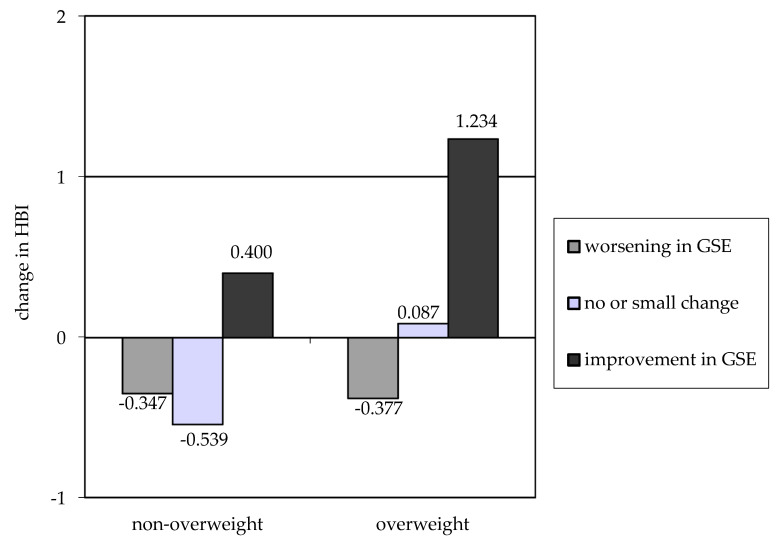
Change in HBI comparing baseline and 3 months’ follow-up after intervention according to BMI group and change in self-efficacy (GSE).

**Figure 3 nutrients-12-02128-f003:**
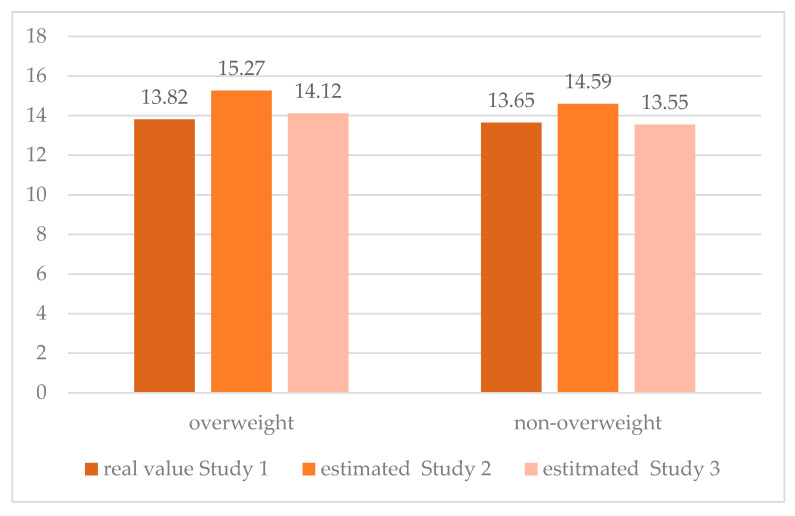
Changes in HBI in three waves of the survey under the Healthy Me program, according to body weight status groups adjusted for type of intervention, initial values of HBI, self-efficacy and interaction body weight status–self-efficacy.

**Table 1 nutrients-12-02128-t001:** Intervention components by type of intervention.

	Intervention Type		
COMPONENTS	Full	Partial	Null
FITNESS BAND			
Objective measurement (steps, heart rate, sleep quality)	X	X	X
MOBILE APP			
Feedback from physical activity telemonitoring (steps, heart rate, distance, sleep quality)	X	X	X
Short messages (facts about a healthy lifestyle)	X	X	
Articles about a healthy lifestyle	X		
Gamification (challenges related to physical activity, nutrition, personal and social competences—individual and group, to be performed alone or in cooperation with family and friends)	X		
**OTHER**			
Workshops at school run by the project coordinator (health education)	X		
Promotion of the intervention theme (physical activity, eating behavior, risk behaviors, personal and social competences) at school and in the local environment—involvement of young people (e.g., preparation by girls of a poster promoting the program and the theme of intervention)	X	X	X
Promotional campaign via Facebook (closed group, competition)	X	X	X

**Table 2 nutrients-12-02128-t002:** Sample characteristics at the baseline.

	Total	Non-Overweight	Overweight
	N ^1^	n (%)	n (%)
Total	1111	843(76.5)	259(23.5)
Type of intervention	n (%)	n (%)	n (%)
Full	597(53.7)	451(76.1)	142(23.9)
Partial	252(22.7)	190(76.3)	59(23.7)
Null	262(23.6)	202(77.7)	58(22.3)
	M ± SD	M ± SD	M ± SD
HBI ^2^	13.69 ± 3.23	13.65 ± 3.19	13.81 ± 3.35
GSE ^3^	34.69 ± 5.33	34.78 ± 5.40	34.37 ± 5.10

^1^ Missing BMI data 0.8% (n = 9); ^2^ HBI—health behavior index; ^3^ GSE—general index of self-efficacy.

**Table 3 nutrients-12-02128-t003:** Health Behavior Index (HBI) change in 3 study periods by the body weight status.

	Total	Overweight	Non-Overweight	*p* ^2^
	M ± S	M ± SD	M ± SD	
Study 1	13.71 ± 3.30	13.81 ± 3.35	13.65 ± 3.19	0.405
Study 2	14.73 ± 4.11	15.12 ± 4.23	14.61 ± 4.07	0.052
Study 3	13.69 ± 3.23	14.16 ± 3.27	13.56 ± 3.29	0.008
*p* ^1^	<0.001	<0.001	<0.001	

^1^ Differences in HBI between 3 study rounds—Kendall’s W test for repeated measures. ^2^ Differences by the body weight status—U Mann–Whitney test for independent groups.

**Table 4 nutrients-12-02128-t004:** Changes in the self-efficacy before and after the Healthy Me program by the body weight status.

Self-Efficacy	Total	Overweight	Non-Overweight	*p* ^3^
General self-efficacy (GSE) ^1^				
Study 1	34.68 ± 5.33	34.36 ± 5.10	34.78 ± 5.40	0.206
Study 3	34.33 ± 5.13	33.99 ± 5.04	34.43 ± 5.16	0.228
*p* ^2^	0.027	0.677	0.030	
Domain of strength				
Study 1	17.38 ± 3.17	17.30 ± 3.07	17.42 ± 3.21	0.451
Study 3	17.16 ± 3.20	17.05 ± 3.30	17.19 ± 3.18	0.696
*p* ^2^	0.023	0.328	0.043	
Domain of perseverance				
Study 1	17.28 ± 3.33	17.06 ± 3.32	17.34 ± 3.33	0.227
Study 3	17.16 ± 3.13	17.02 ± 3.05	17.20 ± 3.15	0.608
*p* ^2^	0.191	0.831	0.165	

^1^ Missing data in GSE 6.6% (Study 1) and 6.2% (Study 3). ^2^ Differences in self-efficacy between 1st and 3rd study rounds—Z Wilcoxon’s test for repeated measures. ^3^ Differences by the body weight status—U Mann–Whitney test for independent groups.

**Table 5 nutrients-12-02128-t005:** Determinants of change in the HBI around the period of Healthy Me intervention identified by generalized linear models.

Predictors	Dependent Variable								
	HBI Change Study 1–Study 2			HBI Change Study 2–Study 3			HBI Change Study 1–Study 3		
	Beta	SE	*p*	Beta	SE	*p*	Beta	SE	*P*
Constant	3.394	0.653	0.000	−1.332	0.908	0.142	4.738	0.899	0.000
Main effect:									
Body weight status									
Overweight	0.412	0.193	0.033	−0.113	0.269	0.674	0.539	0.266	0.043
Non-overweight	Reference category								
Type of intervention									
Full	−0.007	0.203	0.971	0.335	0.282	0.235	−0.357	0.280	0.201
Partial	−0.425	0.241	0.078	0.393	0.336	0.242	−0.833	0.332	0.012
Null (control)	Reference category								
Initial HBI ^1^	−0.397	0.027	0.000	−0.018	0.037	0.618	−0.380	0.037	0.000
Initial GSE ^2^	0.062	0.019	0.001	0.009	0.026	0.725	0.054	0.026	0.040
Interaction:									
Overweight with GSE	0.134	0.034	0.000	0.000	0.048	0.992	0.132	0.048	0.006
Non-overweight with GSE	0.096	0.021	0.000	0.060	0.029	0.038	0.037	0.029	0.198
Scale	6.393	0.291		12.352	0.562		12.106	0.551	
Omnibus test—*p*		0.000			0.455			0.000	

^1^ HBI—health behavior index. ^2^ GSE—general index of self-efficacy.

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
