# Peer review of "Importance of Self-Efficacy in Eating Behavior and Physical Activity Change of Overweight and Non-Overweight Adolescent Girls Participating in Healthy Me: A Lifestyle Intervention with Mobile Technology"

_nutrients, 2020, doi:10.3390/nu12072128_

Round 1
Reviewer 1 Report
The article reported an interesting large-scaled real-world intervention in adolescent girls.
Introduction has been well written and provided sufficient background to the literature.
2.1 - I don't think it's necessary to list down the research questions in this manner. Perhaps it can be incorporated into research objectives and written out in paragraph.
Table 2 to 5 - I have a little concern that the tables may not be very 'friendly' towards non-stats experts. Wonder if authors are able to find another way to present these crucial data.
I have no additional comments besides the above.
Author Response
The authors would like to thank the reviewers for their opinions and the opportunity to revise our paper. The suggestions offered by the reviewers have been immensely helpful. We have edited the manuscript in line with the suggestions and we hope the revised paper addresses all the comments.
Reviewers’ comments:
The article reported an interesting large-scaled real-world intervention in adolescent girls.
Introduction has been well written and provided sufficient background to the literature.
2.1 - I don't think it's necessary to list down the research questions in this manner. Perhaps it can be incorporated into research objectives and written out in paragraph.
As suggested by the Reviewer, the list of questions has been replaced by the short description and the whole paragraph has been moved to the end of the introduction (now in lines 106-113).
Table 2 to 5 - I have a little concern that the tables may not be very 'friendly' towards non-stats experts. Wonder if authors are able to find another way to present these crucial data.
We have made a number of changes to the data presentation in tables. We limited the amount of statistical tests results to “p” values and two tables describing the effect of intervention have been moved to the supplement.
I have no additional comments besides the above.
Reviewer 2 Report
This is an excellent study which has carefully evaluated the Healthy Me study outcomes and placed the results into a context that is important globally to improve the situation with obesity and health. Because of this global applicability, this reviewer believes that the following revisions are needed to the manuscript to make it more scientifically approachable for a wider audience.
- Please shorten the manuscript:
- consider if any of the results tables/figures can be added to the supplemental materials;
- reduce the introduction by at least one paragraph by shortening the theoretical introductory paragraphs
- Make the following changes for clarity in the manuscript:
- Line 68: Remove the word "this" and replace with a phrase that is more specific. It is hard for the reader to understand what "this" is.. is it: "These treatment research studies have"
- line 69: begin with: "Systematic reviews confirm"
- after line 93: Insert a few sentences that describe the Healthy Me project and include a sentence about the HBI and what it is.
- line 112 replace "The authors tried.." with "This study addressed the following.."
- line 405: Please begin the discussion with several direct statements of the results. This paragraph is too general to hold the attention of the reader
- Please reduce the very general nature of the discussion and make it more pointed: you found "x" and it is best interpreted as"Y" within the context of "Z" previous research
- line 504 and 505: change to First and Second
Author Response
The authors would like to thank the reviewers for their opinions and the opportunity to revise our paper. The suggestions offered by the reviewers have been immensely helpful. We have edited the manuscript in line with the suggestions and we hope the revised paper addresses all the comments.
Review 2
This is an excellent study which has carefully evaluated the Healthy Me study outcomes and placed the results into a context that is important globally to improve the situation with obesity and health. Because of this global applicability, this reviewer believes that the following revisions are needed to the manuscript to make it more scientifically approachable for a wider audience.
Please shorten the manuscript:
consider if any of the results tables/figures can be added to the supplemental materials;
We have made a number of changes to the data presentation in tables. We limited the amount of statistical tests results to “p” values and two tables describing the effect of intervention have been moved to the supplement.
reduce the introduction by at least one paragraph by shortening the theoretical introductory paragraphs
We have amended the Introductory text to improve the clarity for the reader (now changes in lines: 66-68; 72-76; 85; 89; 91; 105-115)
Make the following changes for clarity in the manuscript:
Line 68: Remove the word "this" and replace with a phrase that is more specific. It is hard for the reader to understand what "this" is.. is it: "These treatment research studies have"
The sentence has been cancelled when shortening the Introduction section.
line 69: begin with: "Systematic reviews confirm"
The sentence has been cancelled while shortening the Introduction section.
after line 93: Insert a few sentences that describe the Healthy Me project and include a sentence about the HBI and what it is.
We suggest not to add this part of the text in the introduction. The intervention itself as well as the health behavior index are described in detail in the methods section. We kindly ask reviewer to acceptance.
line 112 replace "The authors tried.." with "This study addressed the following.."
Research questions have been shortened to one paragraph and the phrase "the authors tired..." has been removed (now in lines 106-113).
line 405: Please begin the discussion with several direct statements of the results. This paragraph is too general to hold the attention of the reader
As suggested by the Reviewer, we have amended the discussion section.
The first paragraph of the Discussion was changed on (lines 392-398):
In our research, we made an assumption that a change in the level of HBI may be related to a change in the level of personal competence among girls participating in the Healthy Me intervention program which used mobile technologies. Our analyses confirmed this assumption. The change in the HBI was explained by the interaction of the self-efficacy level with body weight status. Although there was no positive change in health behavior among girls without excess weight, girls with excess body weight (overweight or obesity) achieved a better score in the health behavior index in the follow-up study after three months of intervention.
Please reduce the very general nature of the discussion and make it more pointed: you found "x" and it is best interpreted as "Y" within the context of "Z" previous research
We tried to cut the discussion and make it more transparent for the readers (now lines 414-416; 421-432; 436; 431; 445-448; 454-455; 458; 462; 465-467; 469-479; 482-483). The discussion includes a number of references to the results of other studies (references from 36 to 57).
line 504 and 505: change to First and Second
The change was made (lines 482- 483).
Round 2
Reviewer 2 Report
Congratulations to the authors for a careful edit of their manuscript. It reads much better now - and is accessible for a broader audience.This reviewer recommends publication,